# L-Arginine Alleviates the Reduction in Photosynthesis and Antioxidant Activity Induced by Drought Stress in Maize Seedlings

**DOI:** 10.3390/antiox12020482

**Published:** 2023-02-14

**Authors:** Yifei Sun, Feng Miao, Yongchao Wang, Hecheng Liu, Xintao Wang, Hao Wang, Jiameng Guo, Ruixin Shao, Qinghua Yang

**Affiliations:** Henan Engineering Research Center of Crop Chemical Control, Key Laboratory of Regulating and Controlling Crop Growth and Development, Ministry of Education, Henan Agricultural University, Zhengzhou 450046, China

**Keywords:** maize seedling, drought stress, oxidative damage, L-arginine

## Abstract

Maize (*Zea mays* L.) is one of the most important food crops in the world. Drought is currently the most important abiotic factor affecting maize yield. L-arginine has emerged as a nontoxic plant growth regulator that enhances the tolerance of plants to drought. An experiment was conducted to examine the role of L-arginine in alleviating the inhibitory effects of drought on the photosynthetic capacity and activities of antioxidant enzymes when the plants were subjected to drought stress. The results showed that the biomass of maize seedlings decreased significantly under a 20% polyethylene glycol-simulated water deficit compared with the control treatment. However, the exogenous application of L-arginine alleviated the inhibition of maize growth induced by drought stress. Further analysis of the photosynthetic parameters showed that L-arginine partially restored the chloroplasts’ structure under drought stress and increased the contents of chlorophyll, the performance index on an adsorption basis, and Fv/Fm by 151.3%, 105.5%, and 37.1%, respectively. Supplementation with L-arginine also reduced the oxidative damage caused by hydrogen peroxide, malondialdehyde, and superoxide ions by 27.2%, 10.0%, and 31.9%, respectively. Accordingly, the activities of ascorbate peroxidase, catalase, glutathione S-transferase, glutathione reductase, peroxidase, and superoxide dismutase increased by 11.6%, 108.5%, 104.4%, 181.1%, 18.3%, and 46.1%, respectively, under drought. Thus, these findings suggest that L-arginine can improve the drought resistance of maize seedlings by upregulating their rate of photosynthesis and their antioxidant capacity.

## 1. Introduction 

Climate change severely impacts agricultural production, and increasing drought has been identified as one of the most serious meteorological hazards that affect crops [1,2]. Meteorological hazards account for approximately 70% of the total natural hazards globally, and drought accounts for 50% of the global meteorological hazards [3,4]. Drought caused an average annual global economic loss of USD 17.33 billion from 1980 to 2009, which increased to USD 23.125 billion in 2010–2017, far exceeding the losses caused by other meteorological hazards [5,6]. China is among the countries that experience severe and frequent droughts. The government’s emphasis on water projects has reduced the impact of drought on crops in China, but some areas, particularly water-deficient areas, still lack access to irrigation [7]. Improving the drought resistance of crops could alleviate the drought problem. 

The effects of drought stress on plant growth vary with the plant’s growth intensity and growth stage, and different plants have been reported to have reduced drought resistance at the seedling stage [8,9]. Drought directly leads to a lack of water in crops, which results in various interrelated responses that adversely affect plant growth. Studies have shown that drought stress is lethal to underdeveloped seedling roots [10]. The lack of water in the plant leads to reduced plant pigment content, homeostasis imbalance, weakened transpiration, stomatal closure, cell enlargement and reduction, and canopy reduction, which eventually cause plant death [11]. In maize (*Zea mays* L.), drought at the seedling stage leads to restricted plant growth, yield, and quality, which have been recently reported to occur widely worldwide. Thus, it is important to develop mechanisms to alleviate drought stress at the maize seedling stage.

The effect of drought on maize seedlings is partly due to the excessive accumulation of reactive oxygen species (ROS) in plants, which can cause serious oxidative damage to the plant’s organelles [12,13,14]. ROS molecules, including hydrogen peroxide (H_2_O_2_), the superoxide ion (O_2_^•−^), and other oxygen-containing molecules, interact with the membrane system and destroy the existing macromolecules in the cell. An excessive accumulation of ROS can cause oxidative damage to the electron transport chain; enhance lipid peroxidation in the chloroplasts and mitochondria; and inactivate enzymes, proteins, and nucleic acids, ultimately reducing photosynthesis and the yield of crops [15]. Additionally, the accumulation of osmotic fluid can effectively alleviate drought stress and prevent oxidative damage to plants [16,17]. Soluble sugars, prolines, and proteins are the most common osmotic lysates, and their interaction reduces the permeability of cell membranes under mild water scarcity, thereby reducing the maintenance of the water balance at the cellular level in crops under drought stress.

Chemical crop regulation is widely used in agricultural production as a new and efficient cultivation technology. It involves using plant growth regulators to regulate the physiological processes of a crop’s growth and development through the endogenous plant hormone system. The technology can also enhance drought resistance, increase yield, and improve the quality of crops. Exogenous L-arginine is a plant growth regulator and is one of the most versatile amino acids with the highest nitrogen-to-carbon ratio. It is a precursor in the biosynthesis of polyamines (PAs) and signaling molecules, such as nitric oxide (NO) [18,19]. A study on horticultural crops showed that treatment with L-arginine increased the contents of PAs and NO in plants, and improved the resistance of horticultural crops to high temperatures, cold damage, diseases, and other stresses [20]. The positive role of arginine in plant stress responses has been the focus of much research. A study on white button mushrooms (*Agaricus bisporus*) indicated that treatment with 10 mM of L-arginine could maintain the quality of these mushrooms by extending their shelf life [21]. Another study suggested that treatment with arginine attenuated chilling injury in pomegranate (*Punica granatum* L.) fruit during cold storage by enhancing the activity of its antioxidant system, which could also partially maintain the nutraceutical properties of pomegranate fruit [22]. Pretreatment with L-arginine also enhanced the drought resistance of sunflower (*Helianthus annuus* L.) plants, increasing their content of PAs [23]. The positive effects of L-arginine have also been reported in other plants [24,25]. Despite these reports, the function of L-arginine in improving plants’ stress resistance has been only partially verified, and there are few reports on whether exogenous L-arginine enhances the drought resistance of maize seedlings under drought conditions.

Therefore, the purpose of this study was to explore the role and mechanisms of L-arginine in improving the drought tolerance of maize seedlings. ROS homeostasis and photosynthesis, including the chloroplast structure; photosynthetic pigments; and the photosynthetic performance of maize seedlings treated with L-arginine under drought stress were determined. The findings of this study contribute to elucidating the biological function of L-arginine in building resilience against drought stress. 

## 2. Materials and Methods

### 2.1. Plant Material 

The maize variety Xianyu 335, which was provided by Shandong Denghai Xianfeng Seed Industry Co., Ltd. (Shenzhen, China), was used for this study. Uniformly sized seeds capable of producing even and strong seedlings were selected. The seeds were surface sterilized with sodium hypochlorite (15%, *w*/*v*) for 15 min and rinsed several times with distilled water. Thereafter, the sterilized seeds were placed on a tray and kept at 25 °C in the dark for accelerated germination. After the seedlings had grown to one leaf and one heart, 12 robust individuals with uniform growth and no mechanical damage were transplanted to plastic pots (10 cm wide, 30 cm long, and 20 cm high) for hydroponic cultivation. The pots were irrigated with half-strength Hoagland solution once every two days. After the second leaf of the maize seedlings had fully expanded, the pots were irrigated with full-strength Hoagland solution. The Hoagland solution was replaced once every two days. This cultivation process was conducted in an artificial-climate incubator (Zhejiang Top Yunnong Technology Co., Ltd., Hangzhou, China), with day/night temperatures of 25 °C/18 °C, a light intensity of 300 μmol·m^−2^·s^−1^, and a relative humidity of 70 ± 5%. 

### 2.2. Experimental Design 

In our preliminary experiment, one week after transplanting the maize seedlings into plastic pots, a series of L-arginine concentrations were imposed: 0, 50, 150, and 300 μmol·L^−1^ (Appendix A). We found that the growth of maize seedlings treated with 150 μM L-arginine was best among these four treatments (Appendix A). To evaluate the regulatory effect of 150 μmol L^−1^ of L-arginine on photosynthesis and antioxidant activity in maize under drought conditions, we subjected the maize seedlings to five treatments on the 7th day after transplantation when the seedlings had formed three leaves and one heart. N^γ^-nitro-L-arginine methyl ester (L-NAME) can reduce the concentration of arginine-induced endogenous NO in plants [26]. Therefore, these treatments included CK (control treatment), NA (CK + 25 μmol L^−1^ L-NAME), WD (20% polyethylene glycol (PEG)), WD + LA (20% PEG + 150 μmol L^−1^ L-arginine), and WD + NA (20% PEG + 25 μmol L^−1^ L-NAME). The plants receiving 150 μmol L^−1^ of L-arginine (Eke Biotechnology Co., LTD., purity 99%) and 25 μmol·L^−1^ of an L-arginine inhibitor (L-NAME, Beijing Yanbang Technology Co., Ltd., Beijing, China, purity 99%) were first pretreated in the nutrient solution before the experiment. PEG-6000 was used to simulate drought stress three days after the treatments had been administered.

The dynamic changes in the plant growth index and photosynthetic performance were measured on the first, second, and third days after the treatment. The top fully expanded leaves were also collected on the first, second, and third days after the treatment and freeze-dried immediately in liquid nitrogen, followed by storage at −80 °C for analyses of the physiological indices. The treatments were conducted in triplicate, with three plants per replicate.

### 2.3. Vegetative Growth and Relative Leaf Water Content

After 3 days of drought stress, the seedlings were collected and washed with tap water, after which, the fresh and dry weights of their roots and shoots were determined. Immediately after measurement of the fresh weight, the roots and shoots were placed in separate paper bags and oven-dried at 105 °C for 15 min, followed by 75 °C for 3 days. 

Each leaf of the corn seedlings was measured with a ruler. The leaves with the longest lengths had the widest widths. The leaf area was calculated as follows: Leaf area = Leaf length × Leaf width × 0.75.

The fresh weight of fully expanded leaves of healthy plants at the same growth stage was measured. After that, the leaves were immersed in distilled water for 12 h to ensure full saturation before measuring their saturated water content. The samples were then dried at 105 °C for 30 min for degreening, followed by drying to a constant weight at 80 °C. The relative water content (RWC) of the leaves was calculated as described by Shao et al. as follows [27]:RWC = (fresh weight − dry weight)/(saturated fresh weight − dry weight) × 100%.

### 2.4. Arginine Content 

A double antibody one-step sandwich ELISA kit was used to determine the content of arginine. Briefly, the plate’s microwells were precoated with an L-arginine antibody, and the specimens, standards, and the horseradish peroxidase-labeled detection antibodies were added. The plates were then incubated for 60 min in a 37 °C incubator and washed thoroughly. The plates were then stained with a 3,3′,5,5′-tetramethylbenzidine substrate, which turns blue when catalyzed by peroxidase and yellow in the presence of an acid. The depth of the color positively correlated with the L-arginine content of the sample. The absorbance was measured at 450 nm to calculate the concentration of each sample.

### 2.5. Ultrastructure of the Chloroplasts 

For analysis of the chloroplasts’ ultrastructure, leaves without midribs were fixed in a Na phosphate buffer (SPB, 0.2 M, pH 7.2) that contained glutaraldehyde (4%, *v*/*v*) for 1 day at 4 °C and post-fixed in 1% osmium tetroxide for 120 min. The samples were then washed three times with SPB (15 min each) and dehydrated with different concentrations of ethanol (30%, 50%, 70%, 80%, 90%, 95%, and 100%) for 15 min each. The samples were again dehydrated twice with anhydrous ethanol for 20 min each. Thereafter, the samples were inoculated with a mixture that contained absolute acetone and linear resin at ratios of 1:1 and 1:3 for 1 h and 3 h, respectively. The samples were then embedded in pure resin overnight, heated at 70 °C for 9 h, and cut into ultrathin slices using an ultramicrotome, which was followed by staining with lead citrate and uranyl acetate. The stained samples were observed under a transmission electron microscope (Zeiss, Gottingen, Germany).

### 2.6. Photosynthetic Leaf Pigment

The chlorophyll (Chl) content was determined as described by Zhuang et al. [28]. Briefly, 0.1 g of the leaf samples was weighed in glass test tubes that contained 80% acetone and stored in the dark until the samples had become completely discolored. The sample extract was centrifuged, and the absorbance was measured at 470 nm, 645 nm, 652 nm, and 663 nm using a Lambda 25 UV/VIS spectrophotometer (PerkinElmer, Waltham, MA, USA). 

### 2.7. Leaf SPAD and Chlorophyll Fluorescence Parameters

The SPAD was measured using a Chl analyzer. A value was first measured in the middle of the leaf length to avoid the midrib, and the center line of the probe head was aligned at the midpoint of the leaf’s midrib and the edge of the leaf. Another value was measured at 3 cm around the middle of the leaf’s length, and the three SPAD values were measured. The average represented the SPAD value of this leaf. After 20 min of dark adaptation, the maximum photochemical quantum yield (Fv/Fm) and performance index (PI_ABS_) were determined using a handheld fluorometer (Hansatech, King’s Lynn, UK). PI_ABS_ was calculated according to the kinetic parameters of Chl fluorescence induction as described by Kumar et al. [29].

### 2.8. Oxidative Stress Markers

To determine the contents of H_2_O_2_, O_2_^•−^, and malondialdehyde (MDA), we pulverized the frozen samples (0.1 g) into powder by grinding them in liquid nitrogen. The absorbance was then measured at 405, 550, 530, and 520 nm using a 100 mM phosphate buffer (900 µL, pH 7.4) and the specific kits for H_2_O_2_ (A064), O_2_^•−^ (A052), and MDA (a003-3).

### 2.9. Activity of Antioxidant Enzymes 

To assay the antioxidant enzymes, we homogenized 0.5 g of the leaf powder sample with a 100 mM phosphate buffer (900 µL, pH 7.4). Homogenized samples were then centrifuged (12,000× *g* for 15 min at 4 °C), and the supernatant was transferred to new Falcon tubes for analysis of the enzyme activity according to the manufacturer’s instructions (A001-1, A001-1, A123-1, A062-1, A004, A084-3-1, BC0660, and BC0650). The activities of superoxide dismutase (SOD), catalase (CAT), ascorbate peroxidase (APX), glutathione reductase (GR), glutathione S-transferase (GST), and peroxidase (POD) were recorded at wavelengths of 550, 405, 290, 340, 412, 420, 412, and 340 nm, respectively.

### 2.10. Statistical Analysis 

SPSS 22.0 (IBM, Inc., Armonk, NY, USA) was used to analyze the data. The samples were subjected to a one-way analysis of variance (ANOVA), and the treatment means were compared using the LSD (least significant difference) test at *p* ≤ 0.05.

## 3. Results 

### 3.1. L-Arginine Alleviated the Drought-Induced Growth Inhibition of Maize Seedlings 

There was a sharp decrease in dry weight under drought stress. The dry shoot weight of maize seedlings decreased by 62.2% under drought stress, but the dry root weight had no significant difference compared with that of the CK (Figure 1C). Our study found that the dry shoot weight significantly increased by 53.0% under the WD + LA treatment compared with that of WD (Figure 1C). In the NA treatment, the dry shoot and root weights significantly decreased by 59.2% and 45.8%, respectively, compared with the CK (Figure 1C).

Interestingly, compared with the CK treatment, the content of arginine increased by 25.9% under drought stress but decreased significantly by 21.0% under the NA treatment (Figure 1B). The arginine content increased by 11.5% compared with the WD after the exogenous application of L-arginine (Figure 1B). 

### 3.2. Effects of L-Arginine on Photosynthetic Performance under Drought

#### 3.2.1. Leaf Water Conditions and Leaf Area

After 3 days of drought stress, the leaf RWC decreased by 20.8% under WD, while those under WD + LA showed no significant difference compared with the CK. The use of L-NAME also reduced RWC by 22.7% compared with the CK, particularly under drought conditions (Figure 2A). Unlike WD, which decreased the leaf area by 76.5%, LA increased the leaf area by 166.0% (Figure 2B). Additionally, the NA treatment reduced the leaf area by 39.9% compared with that of the CK (Figure 2B).

#### 3.2.2. Ultrastructure of the Chloroplasts 

The chloroplasts formed regular ellipses with stromal lamella layers that were closely arranged and clearly visible under normal conditions and were close to the cell wall, and the outer membrane’s structure was relatively intact (Figure 3A and Appendix A). However, after the drought stress, the chloroplasts became round with a disordered matrix lamella layer, distorted basal particle deformation, and increased osmophilic particles, and they were separated from the cell wall (Figure 3B). However, after the exogenous application of L-arginine, the chloroplast’s morphology returned to a normal elliptical shape, with a normal stromal lamina arrangement and reduced osmophilic granules (Figure 3C). Under the NA treatment, the matrix’s lamella layer was scrambled, and the matrix’s grain was deformed. Larger starch granules appeared within the chloroplasts, and the chloroplasts also separated from the cell wall (Figure 3D).

#### 3.2.3. Photosynthetic Performance

Compared with the CK, the contents of Chl a, Chl b, and total Chl were significantly reduced by 47.3%, 39.7%, and 48.8%, respectively, after 3 days of drought (Figure 4A–C). However, after L-arginine was applied, the photosynthetic pigment contents including Chl a, Chl b, and total Chl significantly increased by 153.2%, 128.9%, and 151.3%, respectively. There was no significant difference between NA and CK, and between WD + NA and WD (Figure 4A–C). The changes in SPAD were consistent with those of the Chl content (Figure 4D). Chl fluorescence parameters, including PI_ABS_ and Fv/Fm, were significantly reduced by 65.7% and 27.4% under WD, respectively (Figure 4E,F). However, after the application of L-arginine, the performance index on an adsorption basis (PI_ABS_) and Fv/Fm increased by 105.5% and 37.1%, respectively, under WD + LA compared with WD (Figure 4E,F). 

### 3.3. Effect of L-Arginine on the Membrane System and Antioxidant Activity under Drought Stress

#### 3.3.1. H_2_O_2_, O_2_^•−^, and MDA

After 3 days of drought treatment, the contents of H_2_O_2_, O_2_^•−^, and MDA had increased in the maize seedling leaves by 50.1%, 11.9%, and 52.2%, respectively, under WD compared with the CK (Figure 5). Applying L-NAME alone increased the contents of H_2_O_2_ and MDA by 6.8% and 16.8%, respectively (Figure 5). However, the contents of H_2_O_2_, O_2_^•−^, and MDA decreased by 27.2%, 10.0%, and 31.9%, respectively, in plants treated with L-arginine compared with WD (Figure 5).

#### 3.3.2. Antioxidant Enzyme Activities

The activities of antioxidant enzymes, such as CAT, GR, SOD, APX, GST, and POD. were determined in maize leaves. The results showed that the activities of CAT, GR, and SOD decreased by 54.4%, 67.5%, and 17.9%, respectively, under WD (Figure 6A,C,D). However, the activities of APX, GST, and POD increased by 2.8%, 17.7%, and 16.8%, respectively, compared with the CK (Figure 6B,E,F). Compared with WD, the activities of CAT, GR, SOD, APX, GST, and POD in WD + LA increased by 108.5%, 181.1%, 46.1%, 11.6%, 104.4%, and 18.3%, respectively (Figure 6). Moreover, L-NAME decreased the activities of GR, APX, and POD by 35.2%, 25.8%, and 6.1%, respectively, compared with the CK (Figure 6A,B,E), but it increased the activities of CAT, SOD, and GST by 11.7%, 13.4%, and 17.6%, respectively (Figure 6C,D,F). 

Interestingly, the arginine content and activities of the six antioxidant enzymes were analyzed by a correlation analysis to elucidate the relationship between the arginine content and these antioxidant enzymes. APX, POD, and GST were positively correlated with the arginine content, while GR, SOD, and CAT were negatively correlated with it (Figure 7). APX had the strongest relationship (R^2^ = 0.7207) with the arginine content, followed by POD (R^2^ = 0.3605), GST (R^2^ = 0.3253) (Figure 6B,E,F), CAT (R^2^=0.1378), SOD (R^2^ = 0.0114), and GR (R^2^ = 0.0077) (Figure 6A,C,D).

## 4. Discussion

Arginine is a direct precursor of NO, urea, functionally diverse amino acids, ornithine, and agmatine [30] and also participates in the biosynthesis of proteins and PAs, osmotic potential, and the vegetative growth of plants [31,32]. PAs are drought-resistant substances that are generally synthesized by the ornithine decarboxylase and arginine decarboxylase pathways [23]. L-arginine has also been found to play a crucial role in stress tolerance in plants [33]. In our study, drought severely inhibited the aboveground growth of maize seedlings, but a significant increase in dry shoot weight was found as a result of the application of exogenous L-arginine (Figure 1). This indicated that arginine is useful for alleviating the inhibition of maize seedling’s growth induced by drought because L-arginine is related to the biosynthesis of signaling molecules [33]. The increased arginine content under drought stress showed that drought stress stimulated arginine production, and exogenous L-arginine application further increased the content of arginine (Figure 1). In a previous study, the exogenous application of arginine to maize plants increased the dry weight of shoots and roots [34]. The reported role of arginine in increasing the growth parameters may be related to the arginine decarboxylase pathways involved in various biological processes [32,35]. In *Arabidopsis thaliana*, the activities of arginine decarboxylase 1 and 2 have been shown to result in drought resistance [36]. The protective effect of arginine on stressed plants may be related to its direct or indirect NO release [37] because NO regulates multiple plant responses to various biotic and abiotic stresses, and alleviates the effects of oxidative stress [38,39]. 

As the site of photosynthesis, the chloroplasts are more sensitive to drought stress, and their structural integrity maintains the normal photosynthetic and biological processes in plants. Therefore, loss of the chloroplast’s structural integrity reduces the photosynthetic function [40]. In this study, the chloroplast structure of maize seedlings was destroyed under WD stress because of the loss of leaf water (Figure 2) and the free radicals, which were produced in large quantities (Figure 3). This resulted in peroxidation and damage to the membrane system, which caused the degradation of Chl and affected the activity of photosynthetic enzymes, thus disabling the photosynthetic apparatus and affecting the normal processes of photosynthesis [41]. However, L-arginine restored the chloroplasts’ shape and reduced the number of osmophilic granules due to the increased leaf RWC (Figure 2 and Figure 3). The restoration of the water state by the exogenous L-arginine treatment under WD stress was consistent with that reported in wheat (*Triticum aestivum* L.) seedlings [42]. In addition, leaf growth determines the light interception capacity of a crop [43,44], and drought stress reduces photosynthesis by decreasing the leaf area and photosynthetic rate per unit of leaf area [45]. The photosynthetic area of leaves can be significantly improved through increased arginine production (Figure 2), which can explain the significant increase in the accumulation of photosynthetic products in maize seedlings after the L-arginine treatment (Figure 1).

Chls are important plant pigments that are used to absorb photons, release electrons, and reflect the light energy utilization of plants [46]. For example, Chl a plays an important role in converting the light energy of absorbed photons into chemical energy under aerobic photosynthesis [47]. Chl b plays an important role in the light-harvesting systems [46]. In our study, drought stress caused a decrease in the Chl content, which increased again after the addition of L-arginine, indicating that L-arginine could maintain the balance between Chl synthesis and degradation under drought. This is consistent with the results of previous studies [48,49]. Meanwhile, Chl fluorescence parameters, such as PI_ABS_ and Fv/Fm, reflect the photosynthetic capability of the entire PSII and the maximal PSⅡefficiency, respectively [50,51]. Drought induced a decrease in PI_ABS_ and Fv/Fm, but L-arginine induced an increase in them (Figure 4), indicating that L-arginine can reduce the inhibitory effect of drought stress on the activity of absorbing photoenergy by the leaves, photochemical activity, and the electron transfer efficiency of PSII’s reaction center, consistent with the findings of a previous study [52].

ROS are easily produced in the chloroplasts since the oxygen concentration is higher in the chloroplast than in other organelles [53]. This reduces the contents of unsaturated fatty acids in the membrane, which leads to membrane protein instability and the loss of function of the membrane’s structure [54,55]. The destructive changes in the chloroplast structure of mesophyll cells that originate from peroxidation of the membrane’s lipid can also be caused by drought [23,56]. In our study, the H_2_O_2_, O_2_^•−^, and MDA contents of leaves were increased in maize seedling after the drought treatment but decreased after the L-arginine treatment (Figure 5). The above results indicated that arginine alleviated oxidative damage in maize under drought stress because arginine could promote the biosynthesis of NO and PAs under drought stress [23,35,57], revealing the role of arginine in improving the plasma membrane’s integrity by reducing or scavenging ROS and MDA. To protect themselves against ROS, plants also have developed numerous antioxidant defense mechanisms that involve various antioxidant enzymes, including CAT, GR, SOD, APX, GST, and POD. When plants are stressed, the antioxidant system is always upregulated to prevent drought-mediated reductions in photosynthesis through the quick elimination of ROS [58,59,60,61]. However, here, the decrease in CAT, GR, and SOD activity (Figure 6) suggested that the maize plants were damaged, because their antioxidant mechanisms did not effectively combat the effects of drought stress [37]. Interestingly, L-arginine reduced oxidative damage with a decrease in the ROS content and products of lipid peroxidation in plants affected by drought (Figure 5), which significantly upregulated the antioxidant system. The activities of antioxidant enzymes and the correlations between arginine content with these activities also proved the positive role of arginine in protecting the membrane’s stability, which alleviated the inhibitory effects of drought stress on maize seedlings. This is consistent with previous studies [57,62,63].

## 5. Conclusions

Drought inhibited the growth of maize seedlings by increasing the production of ROS, which reduced photosynthesis, causing lipid peroxidation (Figure 8). Arginine acts as a precursor of molecular substances such as NO and PAs. The addition of exogenous arginine (L-arginine) could prevent the inhibition of photosynthesis and alleviate the drought-induced oxidative stress in maize seedlings by significantly reducing the accumulation of ROS and lipid peroxidation to maintain the membrane’s functions. Moreover, supplementation with L-arginine improved the antioxidant enzyme activity, demonstrating its beneficial role in preventing drought-induced oxidative damage in maize seedlings. Our study confirmed the role of L-arginine in alleviating drought stress-induced photosynthesis and the reduction in the antioxidant activity of maize seedlings. These findings will have important implications to help humans manage drought stress. In the future, we will study the metabolism of arginine and the genes related to the arginine decarboxylase pathways in drought-stressed maize plants in more detail.

## Figures and Tables

**Figure 1 antioxidants-12-00482-f001:**
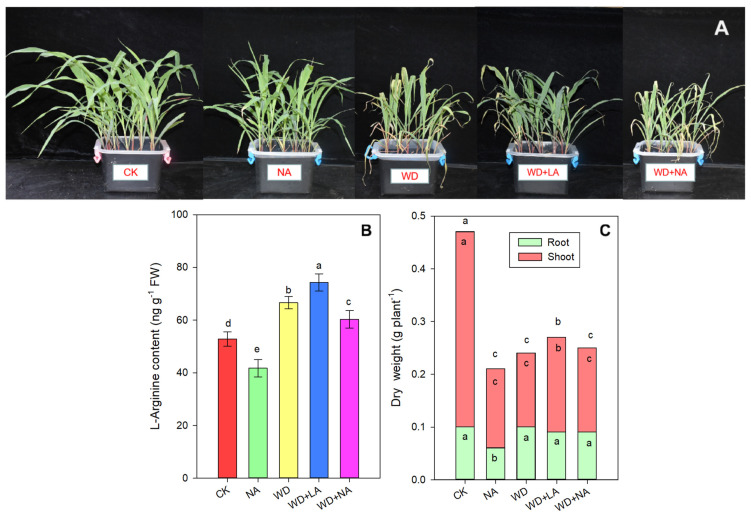
Enhanced growth of maize seedlings induced by supplementation with exogenous L-arginine under drought stress (**A**). Dry weight (**B**) and L-arginine content (**C**) of the maize seedlings under different treatments. CK: control treatment; NA: 25 μmol L^−1^ of L-NAME; WD: 20% PEG; WD + LA: 20% PEG + 150 μmol L^−1^ of L-arginine; WD + NA: 20% PEG + 25 μmol L^−1^ of L-NAME. The results are presented as the means ± standard error (n = 3). Significant differences are represented by lowercase letters (*p* ≤ 0.05) based on the LSD test. Different lowercase letters on the top of the columns in (**B**) indicate significant differences in the biomass of the whole plant (*p* < 0.05). The different lowercase letters in the columns with the same colors in (**C**) indicate significant differences within each treatment (*p* < 0.05), while those above the error bars indicate significant differences among the five treatments (*p* < 0.05). L-NAME, N^γ^-nitro-L-arginine methyl ester; LSD, least significance difference; PEG, polyethylene glycol.

**Figure 2 antioxidants-12-00482-f002:**
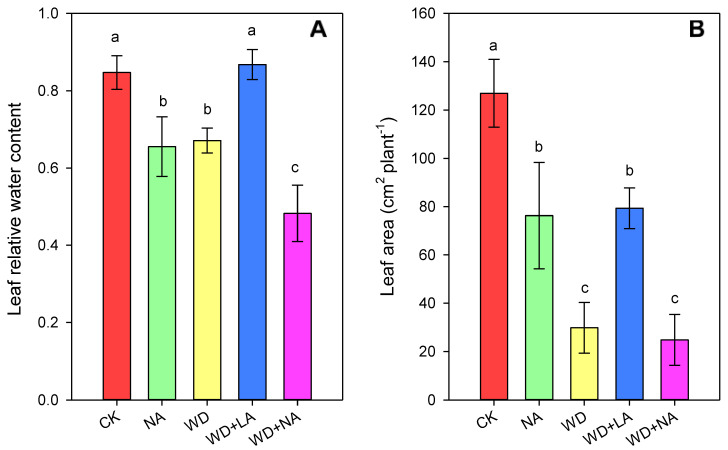
Leaf relative water content (**A**) and leaf area (**B**) of maize seedlings supplemented with L-arginine under drought stress conditions. CK: control treatment; NA: 25 μmol L^−1^ of L-NAME; WD: 20% PEG; WD + LA: 20% PEG + 150 μmol L^−1^ of L-arginine; WD + NA: 20% PEG + 25 μmol L^−1^ of L-NAME. The results are presented as the means ± standard error (*n* = 3). Significant differences are represented by lowercase letters (*p* ≤ 0.05) based on the LSD test. Lowercase letters above the error bars in (**A**,**B**) indicate significant differences among the five treatments (*p* < 0.05). L-NAME, N^γ^-nitro-L-arginine methyl ester; LSD, least significance difference; PEG, polyethylene glycol.

**Figure 3 antioxidants-12-00482-f003:**
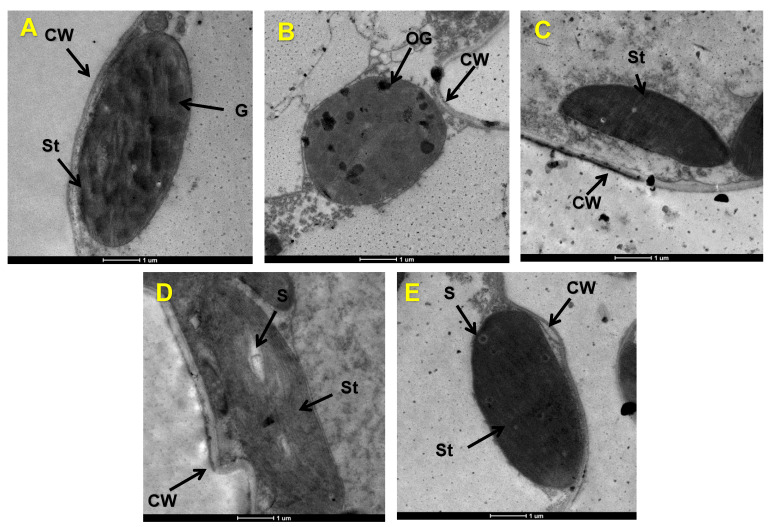
Ultrastructure of the chloroplasts of maize seedlings after exogenous supplementation with L-arginine under drought stress conditions. CW, cell wall; St, stromal thylakoid; OG, osmium granule; S, starch grain. (**A**): control treatment; (**B**): 20% PEG; (**C**): 20% PEG + 150 μmol L^−1^ of L-arginine; (**D**): 25 μmol L^−1^ of L-NAME; (**E**): 20% PEG + 25 μmol L^−1^ of L-NAME. L-NAME, N^γ^-nitro-L-arginine methyl ester; PEG, polyethylene glycol.

**Figure 4 antioxidants-12-00482-f004:**
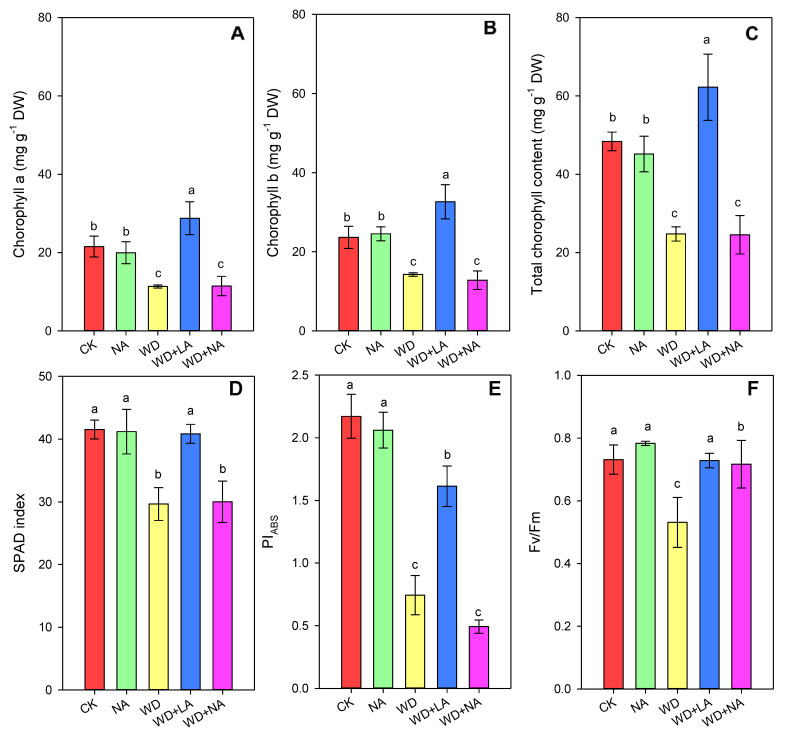
Chlorophyll a (**A**), chlorophyll b (**B**), total chlorophyll (**C**), SPAD index (**D**), PI_ABS_ (**E**), and Fv/Fm (**F**) of maize seedlings under drought stress conditions. CK: control treatment; NA: 25 μmol L^−1^ of L-NAME; WD: 20% PEG; WD + LA: 20% PEG + 150 μmol L^−1^ of L-arginine; WD + NA: 20% PEG + 25 μmol L^−1^ of L-NAME. The results are presented as the means ± standard error (*n* = 3). Significant differences are represented by lowercase letters (*p* ≤ 0.05) based on the LSD test. Lowercase letters above the error bars in (**A**–**F**) indicate significant differences among the five treatments (*p* < 0.05). L-NAME, N^γ^-nitro-L-arginine methyl ester; PEG, polyethylene glycol; PIABS, performance index on an adsorption basis.

**Figure 5 antioxidants-12-00482-f005:**
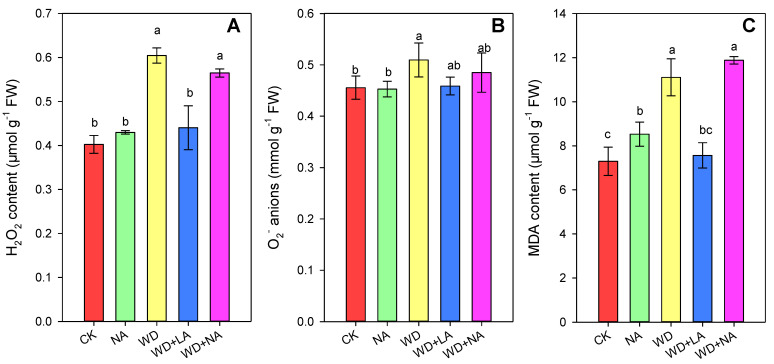
The H_2_O_2_ (**A**), O_2_^•−^ anions (**B**), and MDA contents (**C**) of maize seedlings under drought stress conditions. CK: control treatment; NA: 25 μmol L^−1^ of L-NAME; WD: 20% PEG; WD + LA: 20% PEG + 150 μmol L^−1^ of L-arginine; WD + NA: 20% PEG + 25 μmol L^−1^ of L-NAME. The results are presented as the means ± standard error (*n* = 3). Significant differences are represented by the lowercase letters (*p* ≤ 0.05) based on the LSD test. Lowercase letters above the error bars in (**A**–**C**) indicate significant differences among the five treatments (*p* < 0.05). H_2_O_2_, hydrogen peroxide; L-NAME, N^γ^-nitro-L-arginine methyl ester; LSD, least significance difference; MDA, malondialdehyde; O_2_^•−^, superoxide anion; PEG, polyethylene glycol.

**Figure 6 antioxidants-12-00482-f006:**
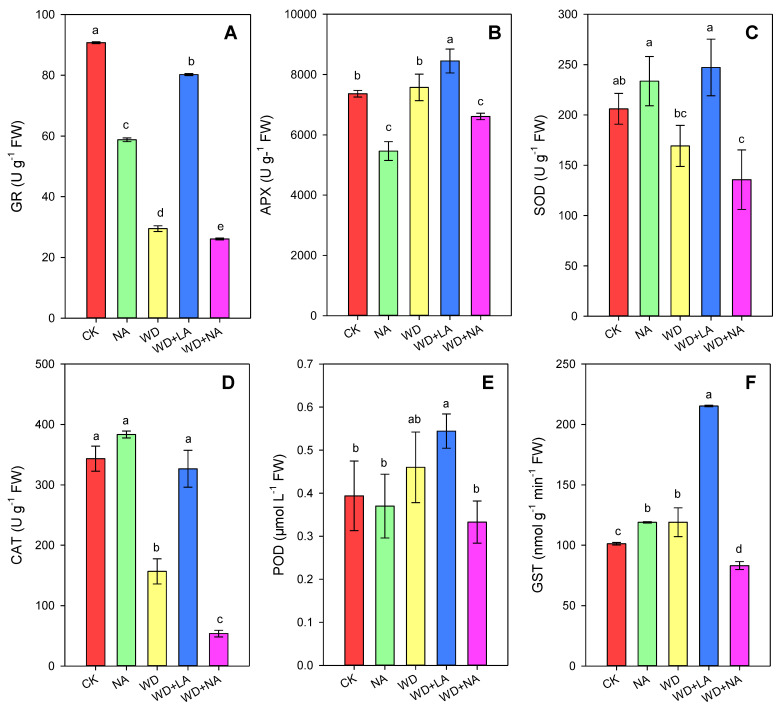
Effects of L-arginine and drought on the antioxidant enzyme activities of glutathione reductase (GR) (**A**), ascorbate peroxidase (APX) (**B**), superoxide dismutase (SOD) (**C**), catalase (CAT) (**D**), peroxidase (POD) (**E**), and glutathione-S-transferase (GST) (**F**) in maize seedlings under different treatments. CK: control treatment; NA: 25 μmol L^−1^ of L-NAME; WD: 20% PEG; WD + LA: 20% PEG + 150 μmol L^−1^ of L-arginine; WD + NA: 20% PEG + 25 μmol L^−1^ of L-NAME. The results are presented as the means ± standard error (*n* = 3). Significant differences are indicated by the lowercase letters (*p* ≤ 0.05) based on the LSD test. Lowercase letters above the error bars in (**A**–**F**) indicate significant differences among the five treatments (*p* < 0.05). L-NAME, N^γ^-nitro-L-arginine methyl ester; LSD, least significant difference.

**Figure 7 antioxidants-12-00482-f007:**
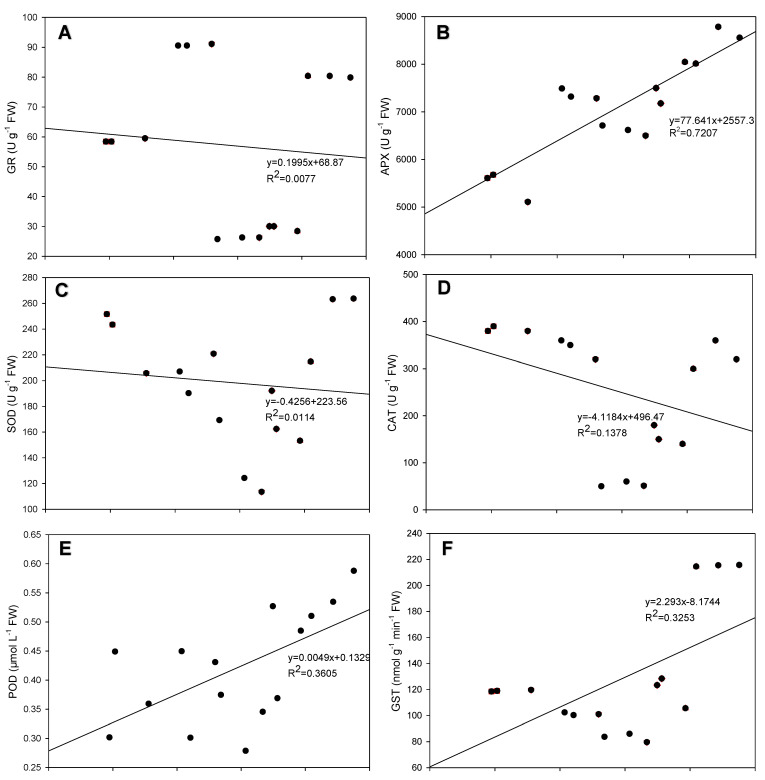
Correlations between arginine content and the activities of glutathione reductase (GR) (**A**), ascorbate peroxidase (APX) (**B**), superoxide dismutase (SOD) (**C**), catalase (CAT) (**D**), peroxidase (POD) (**E**), and glutathione-S-transferase (GST) (**F**).

**Figure 8 antioxidants-12-00482-f008:**
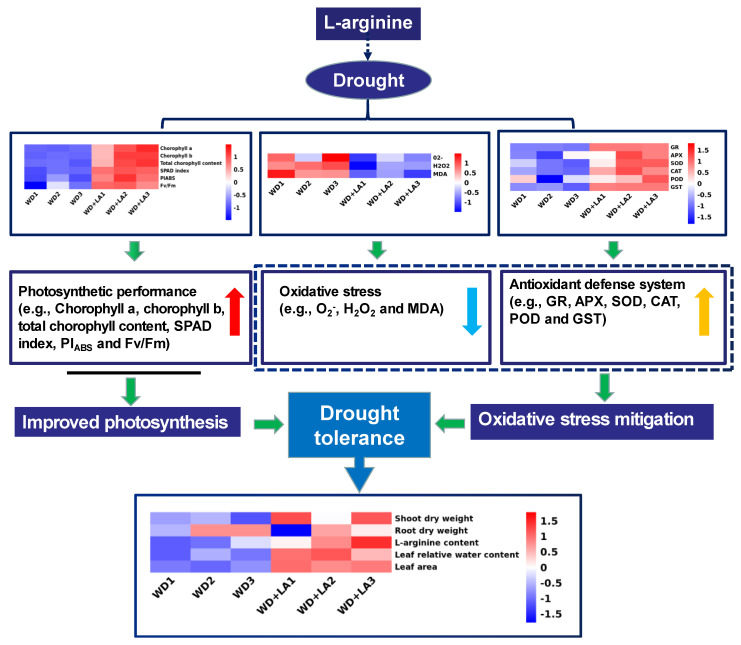
A schematic model describing the regulation of drought stress resistance by L-arginine in maize seedlings. PEG, polyethylene glycol; WD: 20% PEG; WD + LA: 20% PEG + 150 μmol L^−1^ of L-arginine. The red arrow indicates that the photosynthetic performance was enhanced; the blue arrow indicates that oxidative stress was relieved; the yellow arrow indicates that antioxidant enzyme activity levels increased.

## Data Availability

The data presented in this study are available in the graphs provided in the manuscript.

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
