# Peer review of "L-Arginine Alleviates the Reduction in Photosynthesis and Antioxidant Activity Induced by Drought Stress in Maize Seedlings"

_antioxidants, 2023, doi:10.3390/antiox12020482_

Round 1
Reviewer 1 Report
Comments
In general, the submitted manuscript is set out to assess the ameliorative effect of L-arginine on maize seedlings under drought stress. I trust that the paper has the publication potential, but it should be improved in various aspects, as mentioned in the following comments:
Abstract
Line 13: The study not be consice to the one place. It should focus the global issue.
Line 15-17: Should be re-written
The authors should round off the values to minimize the digits in the text.
The studied data is not enough to be published in ANTIOXIDANTS. The authora should also look at the related metabolites and genes to conclude their study.
Author Response
Reviewer 1#
Dear reviewer,
Thank you very much for your valuable comments regarding our manuscript (antioxidants-2148975). We have revised the manuscript in response to your comments. Below are point-by-point details in response to comments.
- Line 13: The study not be consice to the one place. It should focus the global issue.
Response: We agree with your comments and revised the sentence “As one of the main food crops in the world, maize growth is limited by drought stress. ” into “Maize (Zea mays L.) is one of the most important food crops in the world, drought is currently the most important abiotic factor affecting maize yield.” in Line 13 .
- Line 15-17: Should be re-written
Response: We have re-written to be concise and correct. “An experiment was conducted to examine the role of L-arginine in alleviating the inhibitory effects of drought on the photosynthetic capacity and activities of antioxidant enzymes when the plants were subjected to drought stress.” in Line 16 .
- The authors should round off the values to minimize the digits in the text.
Response: We agree with your comments and have reduced the significant digit to 1 digit in the revised manuscript.
- The studied data is not enough to be published in ANTIOXIDANTS. The author should also look at the related metabolites and genes to conclude their study.
Response: Thanks for your comments. Before submitting the paper (antioxidants-2148975), we studied similar literatures published in “Antioxidants” The Role of H2O2-Scavenging Enzymes (Ascorbate Peroxidase and Catalase) in the Tolerance of Lemna minor to Antibiotics: Implications for Phytoremediation. (Gomes et al., 2022); Interactive effects of melatonin and nitrogen improve drought tolerance of maize seedlings by regulating growth and physiochemical attributes. (Ahmad et al., 2022); Grafting enhances pepper water stress tolerance by improving photosynthesis and antioxidant defense systems. (Padilla et al., 2021). There are few reports about related metabolites and genes of L-arginine, so relevant metabolites and genes were not cited in this article. That's what we're trying to do in future. And in Conclusion of the revised manuscript, we added “In the future, we will further study arginine metabolism and genes related with arginine decarboxylase pathways under drought stress in maize”.
Gomes, M. P., Kitamura, R. S. A., Marques, R. Z., Barbato, M. L., & Zámocký, M. (2022). The Role of H2O2-Scavenging Enzymes (Ascorbate Peroxidase and Catalase) in the Tolerance of Lemna minor to Antibiotics: Implications for Phytoremediation. Antioxidants, 11(1), 151.
Ahmad, S., Wang, G. Y., Muhammad, I., Chi, Y. X., Zeeshan, M., Nasar, J., & Zhou, X. B. (2022).
Interactive effects of melatonin and nitrogen improve drought tolerance of maize seedlings by regulating growth and physiochemical attributes. Antioxidants, 11(2), 359.
Padilla, Y. G., Gisbert-Mullor, R., López-Serrano, L., López-Galarza, S., & Calatayud, Á. (2021). Grafting enhances pepper water stress tolerance by improving photosynthesis and antioxidant defense systems. Antioxidants, 10(4), 576.
Again, many thanks to you for your assistance on this manuscript.

Reviewer 2 Report
The submitted manuscript deals with the current issue of the influence of water deficit on juvenile corn plants. The effect of water deficit is reduced by the application of an anti-stress substance, which was L-arginine. It is a natural substance that will certainly find its application in practice. The manuscript is written relatively carefully, but it still needs to be supplemented and edited. It is necessary to explain the abbreviations used in the text. Above all, part of the methodology needs to be supplemented and adjusted. The text states that the plants were grown hydroponically. What was the volume of the containers? How many plants were in the container? Was the solution replaced? Changing the solution is necessary due to the change in its properties due to the action of root exudates and the intake of nutrients by plants. PEG was used as a stressor to simulate water deficit, but PEG induces osmotic stress. The trial scheme should also include a control variant with the application of L-arginine. Is a light intensity of 300 μmol/m2/s sufficient? Leaf area was used as a growth parameter, perhaps it would be appropriate to use the LAI characteristic. The methodology also lacks measurement of SPAD and fluorescence values. As photosynthetic characteristics, I would not only mention the content of pigments and their fluorescence, but the rate of gas exchange. The results are adequate. However, it is necessary to unify the scale on the y-axis for graphs that are related to each other. That's how distorted it is. Chart 7 must be enlarged. The discussion is rather descriptive, please evaluate and analyze your own results. It is necessary to unify the citation of journals, when it is necessary to write it with capital letters.
Author Response
Reviewer #2:
Dear reviewer,
Thank you very much for your valuable comments regarding our manuscript (antioxidants-2148975). We have revised the manuscript in response to your comments. Below are point-by-point details in response to comments.
- It is necessary to explain the abbreviations used in the text.
Response: The abbreviations had been explained in the first time used in the manuscript. For example, we have explained the abbreviation of nitric oxide is NO in Line 80; Abbreviation of water deficit is WD in Line 129;
- The text states that the plants were grown hydroponically. What was the volume of the containers?
Response: We have added the volume (Width 10 cm, length 30 cm, height 20 cm) in Line 113.
- How many plants were in the container?
Response: We have added the plant number 12 in one container (Line 112).
- Was the solution replaced? Changing the solution is necessary due to the change in its properties due to the action of root exudates and the intake of nutrients by plants.
Response: Yes, the solution replaced once every two days. “Hoagland solution were replaced once every two days.” has been added in Line 116.
5.PEG was used as a stressor to simulate water deficit, but PEG induces osmotic stress. The trial scheme should also include a control variant with the application of L-arginine.
Response: Thanks for your valuable suggestions. PEG has been used to simulate water deficit in our studies (Shao et al., 2021; Li et al., 2022). In our preliminary experiment, one week after transplanting into plastic pots, a series of L-Arginine concentrations were imposed 0, 50, 150, 300 (Fig. S1). We found that plant growth of maize seedlings in 150 μM L-Arginine was best among the above four treatments (Fig. S1). So in the next experiment, in order to study the effect of L-Arginine on drought resistance, L-arginine alone was not arranged.
Shao, R., Jia, S., Tang, Y., Zhang, J., Li, H., Li, L., & Zhao, X. (2021). Soil water deficit suppresses development of maize ear by altering metabolism and photosynthesis. Environmental and Experimental Botany, 192, 104651.
Li, H., Tiwari, M., Tang, Y., Wang, L., Yang, S., Long, H., & Shao, R. (2022). Metabolomic and
transcriptomic analyses reveal that sucrose synthase regulates maize pollen viability under heat and drought stress. Ecotoxicology and Environmental Safety, 246, 114191.
Shao, R., Jia, S., Tang, Y., Zhang, J., Li, H., Li, L., & Zhao, X. (2021). Soil water deficit suppresses development of maize ear by altering metabolism and photosynthesis. Environmental and Experimental Botany, 192, 104651.
- Is a light intensity of 300 μmol/m2/s sufficient? Leaf area was used as a growth parameter,perhaps it would be appropriate to use the LAI characteristic.
Response: The light intensity should be 500 μmol/m2/s, it is sufficient for seedlings growth. Here, there are 12 plants in each pots, so we don’t have to calculate LAI.
7.The methodology also lacks measurement of SPAD and fluorescence values.
Response: We have added the measurement of SPAD and fluorescence values in the 2.3
8.As photosynthetic characteristics, I would not only mention the content of pigments and their fluorescence, but the rate of gas exchange. The results are adequate. However, it is necessary to unify the scale on the y-axis for graphs that are related to each other. That's how distorted it is.
Response: Thanks for your valuable comments. We have unified the scale of the y-axis in Fig.4 ABC, please see the revised Fig.4.
- Chart 7 must be enlarged.
Response: The Chart 7 has been enlarged.
10.The discussion is rather descriptive, please evaluate and analyze your own results.
Response: We have improved the Discussion, the details are shown in the manuscript.
- It is necessary to unify the citation of journals, when it is necessary to write it with capital letters.
Response: Thanks much for your advice. We have carefully checked the citation of journals, and revised them.

Round 2
Reviewer 1 Report
The title is still grammatically incorrect!!!
Author Response
Ms. Ref. No.: antioxidants-2148975
Title: L-arginine alleviates the reduction of photosynthesis and antioxidant activity induced by drought stress in maize seedlings
Antioxidants
Dear Editor and reviewer #1,
Thank you very much for your letter regarding our manuscript (antioxidants-2148975). We have revised the manuscript in response to your comments. Below are point by-point details in response to your comments.
- English language and style are fine/minor spell check required.
Response: We have made repeated revisions to your suggestions. And invited the foreign cooperation expert Krishna SV Jagadish to polish the modification.
- The title is still grammatically incorrect!!!
Response: Thanks for your comments, we have made careful changes to the title.We changed “L-arginine alleviates drought stress-induced the reduction of photosynthesis and antioxidant activity of maize seedlings” to “L-arginine alleviates the reduction of photosynthesis and antioxidant activity induced by drought stress in maize seedlings”.
We are now submitting the revised manuscript using the“Track Changes” function, and we had also made other changes in hopes of the manuscript’s acceptance and eventual publication with Antioxidants. For instance, the order 2.3, 2.6 and 2.7 in Materials and Methods had been adjusted, two references had been supplemented in Line 539 and 628, some references had been deleted in Line 478, 487, 494, 499 and 630.
Again, many thanks to you and the reviewers for your assistance on this manuscript.
Yours sincerely,
Pr. Shao
Henan Agricultural University

Reviewer 2 Report
The authors modified the submitted manuscript according to the individual requirements that resulted from the reviews. The changes were justified. Currently, the text is suitable after publication. I am satisfied with the changes and explanations
Author Response
- Ref. No.: antioxidants-2148975
Title: L-arginine alleviates the reduction of photosynthesis and antioxidant activity induced by drought stress in maize seedlings
Antioxidants
Dear Editor and reviewer #2,
Thank you very much for your letter regarding our manuscript (antioxidants-2148975). We have revised the manuscript in response to your comments. Below are point by-point details in response to your comments.
- The authors modified the submitted manuscript according to the individual requirements that resulted from the reviews. The changes were justified. Currently, the text is suitable after publication. I am satisfied with the changes and explanationsary to explain the abbreviations used in the text.
Response: Thank you for your advice and we will continue to work hard.
We are now submitting the revised manuscript using the“Track Changes” function, and we had also made other changes in hopes of the manuscript’s acceptance and eventual publication with Antioxidants. For instance, the order 2.3, 2.6 and 2.7 in Materials and Methods had been adjusted, two references had been supplemented in Line 539 and 628, some references had been deleted in Line 478, 487, 494, 499 and 630.
Again, many thanks to you and the reviewers for your assistance on this manuscript.
Yours sincerely,
Pr. Shao
Henan Agricultural University
